# The Effects of Dietary Crude Protein Level on Ammonia Emissions from Slurry from Lactating Holstein-Friesian Cows as Measured in Open-Circuit Respiration Chambers

**DOI:** 10.3390/ani12101243

**Published:** 2022-05-12

**Authors:** Constantine Bakyusa Katongole, Tianhai Yan

**Affiliations:** Sustainable Agri-Food Sciences Division, Agri-Food and Biosciences Institute, Large Park, Hillsborough, Belfast BT26 6DR, UK; constantine.katongole@gmail.com

**Keywords:** ammonia emission, dietary crude protein, respiration chamber, slurry

## Abstract

**Simple Summary:**

Farmed livestock, particularly dairy cows, are the largest source of ammonia (NH_3_) emissions to the atmosphere in Europe and other parts of the developed world. Generally, more than 80% of the total agricultural NH_3_ emissions in Europe come from manure slurries (mixtures of urine and faeces) with hydrolysis of urea nitrogen (N) in urine, and ammonification of the organic N fraction in faeces as the two main sources of the NH_3_. It is also worth noting that the concentration of these two main sources of NH_3_ emissions from manure slurries (particularly urea N in urine) is positively associated with dietary protein content.

**Abstract:**

The effect of dietary crude protein (CP) level on ammonia (NH_3_) emissions from slurry from lactating Holstein-Friesian cows was studied. Twenty-four-hour total collections of faeces and urine were made from 24 lactating Holstein-Friesian cows fed four total mixed rations containing 141, 151, 177, and 201 g CP/kg DM (6 cows/diet). The collected urine and faeces from each cow were combined to form 2 kg duplicate slurry samples (weight/weight; fresh basis) according to the proportions in which they were excreted. NH_3_ emissions from the slurry samples were measured, during 0–24 and 24–48 h intervals in six open-circuit respiration chambers maintained at two temperatures (8 or 18 °C). NH_3_ emissions for the 0–24 and 0–48 h intervals, as well as the average daily emissions, increased linearly with increasing dietary CP level. Increasing the temperature from 8 to 18 °C positively affected NH_3_ emissions, but only for the 0–24 h interval. In situations where direct measurements are impossible, NH_3_ emissions from slurry can be predicted accurately using equations based on dietary CP level supported by either urinary nitrogen, faeces nitrogen, or both. In summary, increasing dietary CP level linearly increased average daily NH_3_ emissions from slurry, with a 5.4 g increase for each 10 g increase in dietary CP.

## 1. Introduction

Manure slurries (mixtures of urine and faeces) are a significant source of anthropogenic ammonia (NH_3_) emissions to the atmosphere in Europe and other parts of the developed world. In the UK, for example, in 2018, an average of 86% of the total NH_3_ emissions from agriculture came from slurry-related emissions, with 78% for England, 85% for Scotland, 90% for Wales, and 92% for Northern Ireland [1]. The ruminant livestock sector, particularly cattle farming, is the dominant source of NH_3_ emissions from agriculture in Northern Ireland and throughout the UK. The “Making Ammonia Visible Report 2017” produced by the Expert Working Group on Sustainable Agricultural Land Management for Northern Ireland showed that over 70% of NH_3_ emissions in Northern Ireland emanate from cattle production, with the pig and poultry sectors responsible for only 20% of agricultural NH_3_.

NH_3_ is strongly implicated in the damage of ecosystems and biodiversity through its deposition in terrestrial and aquatic environments [2,3], human health respiratory problems through formation of secondary products that contribute to fine particulate matter [4], groundwater contamination with nitrate-N [5], and emissions of the greenhouse gas nitrous oxide (N_2_O) through nitrification–denitrification processes [6,7]. Although NH_3_ is not considered a direct greenhouse gas (because of its short lifetime in the atmosphere), NH_3_ deposition induces N_2_O (a potent greenhouse gas) formation, hence indirectly contributing to the greenhouse effect [7]. Consequently, monitoring and reduction of NH_3_ emissions from livestock farming is a legal requirement in the UK and Europe (revised National Emission Ceilings Directive or Directive 2016/2284/EU; UK National Emissions Ceilings Regulations 2018). Furthermore, storage of manure slurry (a significant source of NH_3_ emissions from livestock) is increasingly becoming necessary due to restrictions of its land application period to only during crop germination and vegetative growth [8].

There are two main sources of NH_3_ emissions from manure slurries: the hydrolysis of urea nitrogen (N) in urine to NH_3_, and the NH_3_ resulting from the ammonification (mineralization) of the organic-N fraction in faeces [9]. Under favourable conditions of warm temperatures, large emission surface areas, and high pH and air velocity, the urinary urea-N is relatively quickly hydrolysed by urease enzyme (which is readily present in faeces [10]) to ammonium-N, resulting in NH_3_ volatilization to the atmosphere (leading to environmental pollution). The hydrolysis starts as soon as urine comes into contact with faeces, and it is completed within 1–2 days [11,12].

Most efforts to reduce release of NH_3_ from manure slurries have focused on improved methods for their storage, treatment, and processing options, as well as their application under field conditions. Reducing the excretion of N in manure through manipulation of animal diets represents another opportunity where effective and economic reductions in subsequent manure NH_3_ emissions could be made [13]. Substantial evidence in the literature supports higher NH_3_ emissions from cattle manure slurries with increasing dietary crude protein (CP) level. For instance, Burgos et al. [14] and Edouard et al. [15] have reported from 2.3- to 4.6-fold increases in NH_3_ emissions associated with manure slurries of dairy cows when dietary CP contents were increased from 15 to 21% DM and 120 to 180 g/kg DM, respectively. The magnitude of N excretion (particularly urinary N), and hence the likelihood of NH_3_ emissions from slurry is highly dependent on dietary CP concentration [16,17,18].

However, most of the studies on NH_3_ volatilization losses from manure slurries completed in different parts of the world have been largely based on the use of micrometeorological and mass N balance techniques. The mathematical solutions for such techniques include some assumptions that must be fulfilled to ensure the results are reliable [19]. These solutions can be quite complex, thus requiring expert users and dedicated software tools. In addition, besides the urinary urea-N concentration and the slurry pH, the hydrolysis of the urinary urea-N content is controlled to the largest extent by climatic factors, particularly air temperature and rainfall [5,20,21]. Thus, there is a need for NH_3_ emission measurements that separate the effect of diet from the effects of geographical and climatic factors. The objective of this study was therefore to determine the effect of increasing dietary CP level on NH_3_ emissions from slurry from lactating Holstein-Friesian cows as measured in open-circuit respiration chambers.

## 2. Materials and Methods

This study was conducted at the Agri-Food and Biosciences Institute (AFBI), Hillsborough (Northern Ireland, UK) under Project Licence Number PPL 2824. All procedures performed in this study involving animals were approved by the Ethical Review Committee of AFBI, and were in accordance with the Animals (Scientific Procedures) Act 1986.

### 2.1. Diets and Feeding of Animals for Slurry Production

The different slurries used in this study were collected from 24 lactating Holstein-Friesian cows of the sister study that evaluated the effect of feeding total mixed ration (TMR) diets containing varying levels of CP on feed intake, nutrient digestibility, milk production, and nitrogen use efficiency [22]. In brief, the 24 cows (between 146 and 200 days in milk, averaging 2.4 parity, 645 kg live weight (SD 57), and 32 kg of milk/cow/day (SD 6) at the time of enrolment in the study) were randomly assigned to four TMR diets containing 141, 151, 177, and 201 g CP/kg DM (Table 1). The TMR diets were prepared from perennial ryegrass (*Lolium perenne* L.) silage and a concentrate mix. The formulation was targeted to produce TMR diets containing 130, 150, 170, and 190 g CP/kg DM in a silage:concentrate ratio of 50:50 (DM basis). However, the actual CP levels were 141, 151, 177, and 201 g CP/kg DM, respectively, in a silage:concentrate ratio of about 48:52 (DM basis).

The cows were offered feed once daily at 09:00 h, and the amounts of feed offered were adjusted daily to obtain approximately 10% orts (as-fed basis) daily throughout 25-day experimental periods. The first 20 days were for adaptation to the TMR treatment diets, followed by a 5-day period of individual feeding (for milk yield, feed intake, and total faeces and urine collection data). The feeding management, feed sampling and laboratory analyses were as described in the sister study [22]. During the 20-day adaptation period, the cows were group-fed their TMR treatment diets. The respective TMR diets were mixed separately (for each treatment) using a feeder wagon (Vari-Cut 12, Redrock, Armagh, Northern Ireland) and tipped into a series of feeders mounted on weigh scales. Individual cow access to the feeders was programmed (cows were able to only access feeders containing their rightful treatment diets) by means of an electronic “neck-tag” identification system (Controlling and Recording Feed Intake (CRFI), BioControl, Rakkestad, Norway).

During the 5-day individual feeding period, the cows were housed in individual tie-stalls (with continuous access to fresh water), and the head of each cow was loosely tied at a halter in the individual tie-stall. The respective TMR diets were mixed separately (for each cow) using a power concrete mixer. The cows were fed their TMR treatment diets once daily at 9:00 a.m., and then any orts (refusals) and spilled feed were collected and weighed on the following day at about 8:00 a.m. (before new feed was offered). The amounts fed were adjusted daily on the basis of the previous day’s intake to obtain approximately 10% orts. The daily TMR intakes were manually recorded.

### 2.2. Faeces and Urine Collection

On Day 1 of each of the 5-day individual feeding periods, each individually confined cow was fitted with a system to facilitate collection of urine separately from faeces. The separation system was comprised of a Velcro patch (vinyl fabric, about 15 cm × 15 cm), which was designed with a hole corresponding to the vulva opening. The patch was glued (EVO-STIK 528 Instant Contact Adhesive, Bostik Ltd., Stafford, UK) to each cow around the perineal region (area under the tail), leaving the vulva opening and anus open. The Velcro patch was supported by two straps (one on either side) that were glued to the rump. For each patch, Velcro tape (loop side) was sewn around the hole corresponding to the vulva, which helped to hold in position a urine collection unit (with hook-side Velcro tape) connected to an 80 mm polyvinyl chloride (PVC) Layflat Hose (for channelling the urine into a plastic collection container).

On the following day, 24 h total collections of faeces and urine were made. To prevent NH_3_ from volatilizing, urine was collected on ice (i.e., after inserting each plastic urine collection container in an ice box). The faeces of each cow were collected in a large plastic collection tray that was placed behind each tie-stall. The 24 h faeces and urine collections were weighed at 08:00 a.m. About 4 and 2 kg aliquots of the faeces and urine, respectively, of each cow were taken and stored in a refrigerator (at 5 °C) for later assessment of NH_3_ emissions from the ensuing slurry. The faeces samples were collected in plastic bags and tied tightly, while urine samples were collected in plastic screw-capped containers.

### 2.3. Slurry Sample Preparation

The refrigerated urine and faeces of each cow were combined to form 2 kg duplicate samples of slurry (weight/weight; fresh basis) according to the proportions in which they were excreted by the respective cows (simulating urine and faeces deposition in concrete floor cattle housing systems). The urine:faeces production ratios averaged 1:2.27, 1:2.15, 1:2.21, and 1:1.65 (weight/weight; fresh basis) for the 141, 151, 177, and 201 g CP/kg DM diets, respectively.

### 2.4. Slurry Allocation to Respiration Chambers and NH_3_ Emission Measurements

Considering that there were only six open-circuit respiration chambers (RC) at AFBI, NH_3_ emission measurements were conducted in eight batches (chamber runs) at intervals of 2–3 days. The RCs were numbered from 1 to 6, and arranged in two rows facing one another (with RC1 facing RC6, RC2 facing RC5, and RC3 facing RC6). On a random basis, RCs 1, 2, and 3 were maintained at 8 °C, while RCs 4, 5, and 6 were maintained at 18 °C. For each of the eight chamber runs, slurry samples of only three randomly selected cows were studied. The duplicate slurry samples from the same cow were randomly assigned to one of two chamber temperatures (8 and 18 °C), while ensuring that slurry samples from the same cow were not assigned to RCs directly facing each other (Figure 1).

To constitute the 2 kg slurry samples, the corresponding urine and faeces quantities were weighed out separately. While inside their randomly assigned RCs, the weighed out quantities of urine and faeces were quickly poured into a plastic bucket (with the RC door closed), stirred for 1 min using a Mac Allister paddle mixer, and thereafter the entire mixture was poured off on a plastic tray (8 × 44 × 75 cm height, width and length, respectively), spreading it in a thin layer to cover the entire surface of the tray.

The six indirect open-circuit RCs used were made with double Perspex (Lucite International, Darwen, UK) walls fitted in aluminium frames and mounted on a profiled plastic floor, with a single unplasticised polyvinyl chloride (uPVC) door. The total volume of 4.86 m^3^ (1.98 m long, 1.46 m wide, and 1.68 m high) for each chamber was ventilated by suction pumps set at 10 Normal Cubic Metres per Hour (NCM/hour), giving approximately 1.6 air changes/hour. Temperature and relative humidity control were achieved with Hitachi air-conditioning units set at either 8 ± 1 °C or 18 ± 1 °C and 50 ± 10%, respectively. The chambers were operated under negative pressure (30 N/m^2^), with exhaust air removed at one position (through a 50 mm (internal diameter) PVC duct) for volume measurement and gas analysis.

Chamber air samples were taken from the 50 mm exhaust ducts through 6 mm polytetrafluoroethylene (PTFE) Teflon tubing by an ADC MGA 3500 gas sampling unit (Analytical Development Co. Ltd., Hoddesdon, UK). NH_3_ concentrations were measured using a LumaSense, INNOVA 1512i Photoacoustic Gas Monitor (Lumasense Technologies A/S, Denmark). Temperature and humidity were measured using a TREND HT/D/2% combined temperature/humidity sensor (TREND Control Systems, Horsham, UK), and atmospheric pressure was measured using a Vaisala PTA 427 digital barometer (Delta-T Devices, Cambridge, UK). Results were collected via TREND IQ4 Controllers, (TREND Control Systems, Horsham, UK) for 2 min from each chamber, and the average values of the final minute were logged into an SQL Database on the secure server on site for post-run analysis, giving 4.3 measurements/chamber per hour. The analysis train was designed to run automatically. Prior to the beginning and at the end of each experimental period, the chambers were tested by comparing the amount of NH_3_ recorded by the ADC MGA3500 gas analyser 3 h after the release of a known quantity of 800 ppm NH_3_ by Mass Flow Controllers (4 L/min) (Cole Palmer, www.colepalmer.co.uk, accessed on 9 May 2022). The recovery was found to be almost 75% in all periods. Samples remained in the chamber for 72 h, with all NH_3_ measurements (in litres) for the final 48 h (in two time intervals of 0–24 and 0–48 h). These measurements accounted for NH_3_ released by 1 m^2^ of slurry spread on the plastic tray placed on the floor of the chambers. NH_3_ emissions from the 2 kg slurry samples were converted to grams of NH_3_ per kg of fresh slurry, and then extrapolated to grams of NH_3_ per cow per day after factoring in the total amount of slurry for each cow per day.

### 2.5. Laboratory Analysises

For each of the 24 refrigerated faeces and urine samples, about 500 g and 200 mL, respectively, were taken for laboratory analysis. A subsample of about 100 g was taken from each of the refrigerated faeces samples and analysed for total N content using wet chemistry analysis. The rest of the faeces sample was oven-dried at 60 °C for 144 h (for DM content determination). After oven-drying, the samples were ground through a 1 mm sieve and analysed for neutral detergent fibre (NDF) and acid detergent fibre (ADF) concentrations [24]. The urine samples were analysed for total N content. The total N concentration in faeces and urine was analysed using a Tecator Kjeldahl Auto 1030 Analyser (Foss Tecator AB, Höganäs, Sweden).

### 2.6. Statistical Analysis

Data were analysed using SAS (version 9.1, SAS Institute, Cary, NC, USA, 2003). Differences in NH_3_ emissions were considered significant when *p* ≤ 0.05, and tendencies towards significant differences when 0.05 ≤ *p* ≤ 0.10. Data were analysed using the PROC MIXED procedure. The interaction of dietary CP level × chamber temperature was initially included in the model as a fixed effect, but was removed because it was not significant in all cases. Therefore, statistical models containing only the main effects of dietary CP level or chamber temperature were fitted. The statistical models had the following forms:Y_ijk_ = μ + P_i_ + cow(cr)_j(i)_ + PN_ik_ + e_ijk_
where Y_ijk_ = NH_3_ emissions per cow, μ = overall mean effect, P_i_ = fixed effect of either dietary CP level or chamber temperature, cow(cr)_j(i)_ = random effect of the cow from which the slurry was collected within the respiration chamber run (order of investigation in the respiration chambers), PN_ik_ = fixed effect of the dietary CP level × respiration chamber number interaction, and e_ijk_ = effect of the residual error.

Selection of the fixed and random effects depended on whether their inclusion in the model resulted in a smaller Akaike information criterion (AIC) and/or made a meaningful improvement in treatment means and standard errors. Polynomial contrasts were used to detect linear and quadratic responses to dietary CP level.

The two-sample *t*-test was used to compare (within dietary CP level or chamber temperature groups) the average NH_3_ emissions measured for the 24–48 and 0–48 h intervals, as well as the average daily NH_3_ emissions with the 0–24 h average NH_3_ emissions. Linear regressions of NH_3_ emissions with dietary CP level, total urinary nitrogen (UN), and faeces nitrogen (FN) were performed using the PROC REG procedure of SAS. Regression coefficients of the regression equations for 8 and 18 °C chamber temperatures were compared.

## 3. Results and Discussion

### 3.1. Urine and Faeces Characteristics

Table 2 provides a summary of the weights and chemical characteristics of the urine and faeces combined to form the slurry samples. The amount of urine increased linearly (*p* < 0.05) with increasing dietary CP level, being higher with the 177 or 201 diets (22.9 and 23.2 kg/day, respectively). As dietary CP level was increased from 141 to 151, 177, and then 201 g/kg DM, the amount of urine increased by 6.4, 45.9, and 47.8%, respectively. Rumen-degradable protein (RDP) concentrations were relatively high in the high-CP diets (Table 1), which might have resulted in excess rumen NH_3_-N being excreted as urea in urine and reduced blood urea-N transfer to the rumen for recycling [25]. When dietary RDP is in excess of the amount required by ruminal microorganisms, the protein is degraded to NH_3_-N, absorbed, metabolized to urea in the liver, and lost in the urine [26] rather than being salvaged and recycled to the rumen. However, the excretion of urea in urine requires water [27], which inevitably leads to higher water intake and, hence, increased urine amount. Others studies [14,28] also reported linear increases in urine amount with increasing dietary CP level.

The amount of faeces voided and the N, DM, NDF, and ADF contents of the faeces did not differ (*p* > 0.05) across dietary CP levels, and averaged 39.2 kg/cow/day and 5.0 g/kg (fresh), 147.2 g/kg, 552.5 g/kg, and 291.0 g/kg DM, respectively (Table 2). The lack of dietary CP level effect on faecal N content is consistent with Van Vliet et al. [29], who observed no significant differences in the faecal N content of cows fed high-protein diets compared with those fed low-protein diets. According to the nutrient digestibility results of the sister study [22], the low-protein diets had the lowest apparent total-tract digestibility of CP, which obviously resulted in increased faecal N excretion for the low-protein diets, thus causing the lack of difference observed.

The amount of total slurry (mixture of urine and faeces) was affected in a linear (*p* < 0.05) manner, showing a tendency (*p* = 0.070) for a quadratic response to increasing dietary CP level. This effect was attributed more to the higher urine amount (23.2 kg/day for the 201 diet compared with 15.7 kg/day for the 141 diet), particularly since the effect of dietary CP level on amount of faeces was not significant (*p* > 0.05). Similarly, Burgos et al. [14] also reported a linear increase in the amount of slurry as dietary CP level was increased from 15 to 17, 19, and then 21% DM.

### 3.2. Dietary CP Level and NH_3_ Emissions from Slurry

The effect of dietary CP level on NH_3_ emissions from slurry from lactating cows by measurement time interval is presented in Table 3. The interaction between dietary CP level and chamber temperature was not significant (*p* > 0.05); thus, only tests on main effects are presented. The average NH_3_ emissions for the 0–24 h interval increased in a linear (*p* < 0.05) manner with increasing dietary CP level, being higher (*p* < 0.05) with the 177 or 201 diets and not different between the 141 and 151 diets. The average NH_3_ emissions for the 24–48 h interval increased in a linear (*p* < 0.05) manner with increasing dietary CP level, being highest (*p* < 0.05) with the 177 or 201 diets. The average NH_3_ emissions for the 0–48 h interval also increased in a linear (*p* < 0.05) manner with increasing dietary CP level, being highest with the 201 diet, intermediate with the 177 diet, and lowest with the 141 and 151 diets. The same pattern was observed for the average daily NH_3_ emissions.

The average NH_3_ emissions for the 0–24, 24–48, and 0–48 h intervals and the average daily NH_3_ emissions were increased by 2.9- to 3.5-fold as dietary CP level was increased from 141 to 201 g/kg DM. This result was expected and corroborates the observation that the principal driver of NH_3_ emission from slurries is dietary CP level or N intake [17,18]. Accordingly, previous studies [13,14,15] reported higher NH_3_ emissions from manure slurries as dietary CP level was increased. Most NH_3_ emissions from slurries are produced from the hydrolysis of urinary urea-N to ammonium (NH_4_^+^) by urease enzyme [30] that is present in faeces [10] from rumen bacteria [31]. High dietary CP levels result in increased rumen NH_3_-N concentration relative to rumen microbial requirements [32], which has been reported to be inversely related to urea-N transfer to the rumen [25] for recycling. Consequently, the total urea-N salvaged and recycled to the rumen decreases as dietary CP level increases, which ends up being excreted in urine rather than being salvaged and reutilized. Therefore, because increasing dietary CP level increases urinary urea-N content, an increase in dietary CP level leads to an increased potential for NH_3_ volatilisation [33,34]. For this reason, Arriaga et al. [35] observed a 36.5% reduction in NH_3_ concentration on tie-stall floors, and without compromising milk nitrogen use efficiency when dietary CP content was reduced from 17 to 14% DM in mid- to late-lactating cows.

Unsurprisingly, the difference between diet 177 and diet 201 for the 0–24 h average NH_3_ emissions was significant, but not significant for the 24–48 h interval (Table 3). Previous studies [36,37] have shown that the urease hydrolysis of urea-N in slurry to NH_4_^+^ (and thus NH_3_ volatilization) is complete within approximately 20 h after the urine and faeces come into contact. This rapid hydrolysis of urea-N in slurry to NH_4_^+^ decreases the available NH_4_^+^ to be converted to NH_3_ [38], hence causing the levelling off of NH_3_ emissions for those diets.

Results of the two-sample *t*-tests (within each dietary CP level group; Figure 2) indicated that the average NH_3_ emissions for the 0–24 h intervals were higher (*p* < 0.05) than for the 24–48 h intervals across the four dietary CP levels. Chantigny et al. [39] reported higher NH_3_ emission rates from slurry within the first 12 h following surface application in the field. Moal et al. [40] reported that an average of 75% of the total NH_3_ emissions occurred within the first 15 h following the surface spreading of pig and cattle slurry, while Carozzi et al. [41] reported that the last high peak of NH_3_ emissions following the surface spreading of dairy slurry occurred after 24 h before gradually decreasing. According to Sommer and Hutchings [20] and Sommer et al. [42], NH_3_ emission rate is usually highest immediately after slurry application, partly due to the high initial concentration of ammonium-N (NH_4_^+^-N). Christensen et al. [43] observed that approximately 70% of the N in slurry is present as NH_4_^+^, which is lost as NH_3_ due to volatilization. However, the rate normally falls rapidly as the concentration of NH_4_^+^ decreases after the high initial NH_3_ emission [44], levelling off within the first 30 h [39]. This explains why the average NH_3_ emissions for the 0–24 h interval was significantly higher (*p* < 0.05) than for the 24–48 h interval.

The two-sample *t*-tests between the 0–24 and 0–48 h intervals showed that the average NH_3_ emissions for the 0–48 h intervals (i.e., total cumulative NH_3_ emissions for the entire study period) were higher (*p* < 0.05) than for the 0–24 h intervals for the 151, 177, and 201 diets. For the 141 diet, the average NH_3_ emissions for the 0–48 h interval tended (*p* = 0.092) to be higher. The average NH_3_ emissions observed in this study for the 0–24 h interval represented 68, 68, 67, and 67% of the total cumulative NH_3_ emissions (0–48 h average emissions) from the slurries from cows fed diets containing 141, 151, 177, and 201 g CP/kg DM, respectively. Sommer and Hutchings [20] and Sommer et al. [42] reported a value of close to 50% following the field application of slurry. We attributed much of this disparity to the different methodologies and conditions in which these studies were conducted with respect to measurement technique (open-circuit respiration chambers vs. micrometeorological methods), instruments used, location (enclosure vs. open air), slurry characteristics, and climatic conditions (controlled vs. uncontrolled environment). Each of these factors can influence NH_3_ emission estimates [19,30].

### 3.3. Chamber Temperature and NH_3_ Emissions from Slurry

The fffect of chamber temperature on NH_3_ emissions from slurry from lactating cows by measurement time interval is presented in Table 4. The average NH_3_ emissions for the 0–24 h interval were greater (*p* < 0.05) for 18 °C than for 8 °C. Maintaining the chamber temperature at 18 °C resulted in about 35% greater average NH_3_ emissions for the 0–24 h interval, compared with 8 °C. The effect of chamber temperature on the average daily NH_3_ emissions was not significant (*p* > 0.05), and averaged 31 g of NH_3_/cow/day. The same pattern was observed for the average NH_3_ emissions for the 0–48 h interval, averaging 62 g of NH_3_/cow. According to Sommer et al. [42], the relationship between NH_3_ emissions from slurry and measurement temperature is often simply a reflection of the warming effect (increased temperature), which displaces the NH_3_ and NH_4_^+^ equilibrium to NH_3_ [45], and also increases urea activity.

The absence of differences in average NH_3_ emissions between the 8 and 18 °C for the 0–48 h interval and the average daily NH_3_ emissions could be attributed to the fact that in general, NH_3_ emissions from manure slurries depend on the concentration of free NH_3_ in aqueous solutions [46]. As described earlier in this paper, the rate of NH_3_ production from manure slurries is determined by the availability of NH_4_^+^, which dissociates to form NH_3_ and H^+^ [46]. The rate normally falls rapidly as the concentration of NH_4_^+^ decreases [44], levelling off within the first 30 h [39]. This also explains why 8 °C yielded significantly higher (*p* < 0.05) average NH_3_ emissions for the 24–48 h interval compared with 18 °C.

The two-sample *t*-tests (within each chamber temperature group; Figure 3) comparing the average NH_3_ emissions for the 0–24 h interval with the 24–48 h interval and the average daily NH_3_ emissions produced mixed results. Within the 8 °C chamber temperature group, the average NH_3_ emissions for the 0–24 and 24–48 h intervals as well as the average daily NH_3_ emissions were similar (*p* > 0.05). However, a different pattern was observed for the 18 °C chamber temperature group, in which the average NH_3_ emissions for the 0–24 h interval were higher (*p* < 0.05) than for the 24–48 h interval and the average daily NH_3_ emissions. Similarly, the two-sample *t*-tests between the 0–24 and 0–48 h intervals also produced mixed results. Within the 8 °C chamber temperature group, the average NH_3_ emissions for the 0–24 h interval were lower (*p* < 0.05) than for the 0–48 h interval. However, a different pattern was observed for the 18 °C chamber temperature group, in which the average NH_3_ emissions for the 0–24 and 0–48 h intervals were not different (*p* > 0.05).

These mixed results might be attributable to the effect of the interaction between pH and temperature on urease activity and the dissociation of NH_4_^+^. As described earlier in this paper, the dissociation of NH_4_^+^ is dependent on manure properties, including temperature and pH among others [46]. Increases in temperature and pH of the liquid manure increase the NH_4_^+^ dissociation and thus NH_3_ volatilization [46,47]. However, Liu et al. [45] observed that temperature within a range of 10–25 °C had little if any effect on NH_4_^+^ dissociation for a normal pH range of 6.5–9. This could explain why the average NH_3_ emissions for the 0–24 and 0–48 h intervals were not different for the 18 °C chamber temperature group.

### 3.4. Relationships of NH_3_ Emissions from Slurry and Dietary CP Level, Total Urinary N, and Faeces N

The linear regression equations for predicting NH_3_ emissions from slurry from lactating cows based on dietary CP level, total urinary nitrogen (UN), and faeces N (FN) are presented in Table 5. The numbers in parentheses are the standard errors of the coefficient estimates. Slopes of the regression equations for predicting NH_3_ emissions were not different (*p* > 0.05) between the two chamber temperatures, so overall relationships (regardless of chamber temperature) are presented. Using dietary CP level, total UN, or FN contents as the only predictors of NH_3_ emissions from slurry indicated positive relationships (Equations (1)–(3)). The best single independent variable was dietary CP level (R^2^ = 0.67), closely followed by total UN content (R^2^ = 0.62). The relationship between NH_3_ emissions and FN was weak, which was consistent with the fact that N in faeces is less susceptible to NH_3_ volatilization losses than urea-N in urine. The addition of either total UN or FN as supporting predictors to dietary CP considerably increased the prediction accuracy to 0.72 and 0.71, respectively (Equations (4) and (5)). Combining the three variables (dietary CP level, total UN, and FN) only marginally increased the prediction accuracy to 0.73.

## 4. Conclusions

The results showed that increasing dietary CP level linearly increased the average daily NH_3_ emissions from slurry and emissions for the 0–24, 24–48, and 0–48 h intervals. Each 10 g increase in dietary CP level was associated with a 5.4 g increase in the average daily NH_3_ emissions. Regardless of dietary CP level, over 65% of the total NH_3_ emissions from slurry occurred within the first 24 h of urine coming into contact with faeces. Increasing the chamber temperature from 8 to 18 °C also increased NH_3_ emissions from slurry, but only for the 0–24 h interval. There were strong relationships of NH_3_ emissions from slurry with dietary CP level and with total urinary N content. The most accurate equations for predicting NH_3_ emissions from slurry were those based on dietary CP level supported by either the contents of urinary nitrogen, faeces nitrogen, or both.

## Figures and Tables

**Figure 1 animals-12-01243-f001:**
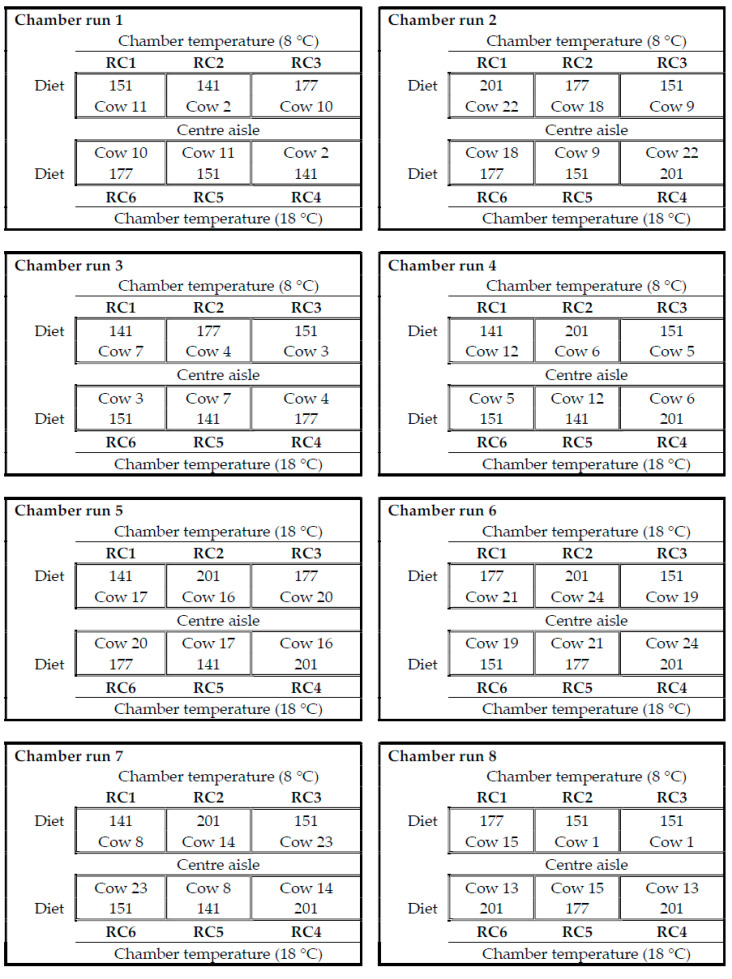
Layout of the allocation of slurry sub-samples of the 24 cows (from cow 1 to cow 24) to the 6 open-circuit respiration chambers (from RC1 to RC6).

**Figure 2 animals-12-01243-f002:**
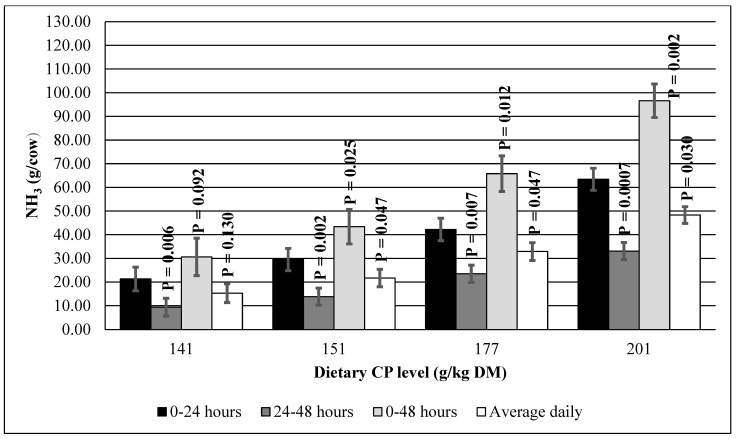
NH_3_ emissions from slurry from lactating cows fed at 4 dietary CP levels (141, 151, 177, and 201 g CP/kg DM) measured at different time intervals (0–24, 24–48, and 0–48 h) in open-circuit respiration chambers. The figure presents 24–48 and 0–48 h averages, as well as daily averages compared with the 0–24 h averages (within dietary CP levels) using the two-sample *t*-test. Vertical bars represent standard errors of the means.

**Figure 3 animals-12-01243-f003:**
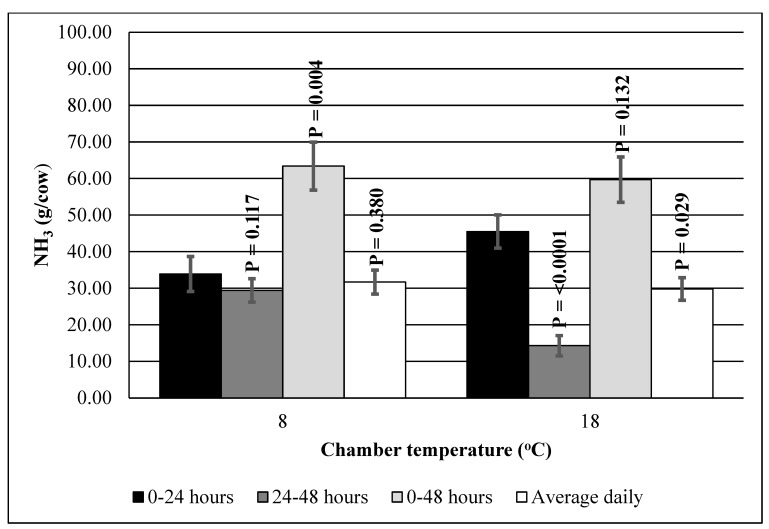
NH_3_ emissions from slurry from lactating cows measured at 2 temperatures (8 and 18 °C) at different time intervals (0–24, 24–48, and 0–48 h) in open-circuit respiration chambers. The figure presents 24–48 and 0–48 h averages, as well as daily averages compared with the 0–24 h averages (within each chamber temperature) using the two-sample *t*-test. Vertical bars represent standard errors of the means.

**Table 1 animals-12-01243-t001:** Composition of the TMR treatment diets.

	Dietary CP Content (Formulated), g/kg DM			
	141	151	177	201
Ingredient, % of TMR DM				
Ryegrass silage	47.3	47.9	47.7	47.5
Concentrate mix	52.7	52.1	52.3	52.5
Chemical composition ^1^				
Dry matter (DM), g/kg	507	513	510	508
Crude protein (CP), g/kg DM	141	151	177	201
Rumen-degradable protein (RDP) ^2^, g/kg CP	103	109	127	144
Rumen-undegradable protein (RUP) ^2^, g/kg CP	39	41	49	56
Neutral detergent fibre (NDF), g/kg DM	411	408	399	388
Acid detergent fibre (ADF), g/kg DM	218	218	214	209
Water-soluble carbohydrates) WSC, g/kg DM	81	82	86	90
Ash, g/kg DM	78	78	80	81

^1^ Calculated as formulated from the analysed chemical compositions of the individual ingredients; ^2^ Feed into milk (FiM) system according to Thomas [23].

**Table 2 animals-12-01243-t002:** Weight and chemical characteristics of the urine and faeces combined to form the slurry samples.

	Dietary CP Level, g/kg DM					*p*-Value		
	141	151	177	201	SEM	Diet	Linear	Quadratic
Urine								
Weight, kg/cow/day	16.7 ^b^	16.8 ^b^	21.6 ^ab^	23.2 ^a^	1.85	0.039	0.006	0.805
Total urine N, g/L	8.0 ^b^	8.5 ^b^	9.5 ^b^	12.0 ^a^	0.53	0.0001	<0.0001	0.179
Faeces								
Weight, kg/cow/day	34.9	36.3	46.3	39.4	4.66	0.343	0.275	0.180
Total faeces N, g/kg (fresh)	4.9	4.8	5.1	5.2	0.36	0.812	0.393	0.942
DM, g/kg	149.5	145.7	140.8	152.9	7.24	0.654	0.770	0.242
NDF, g/kg DM	545.7	559.6	570.2	534.3	10.1	0.108	0.412	0.082
ADF, g/kg DM	286.3	299.1	298.7	279.9	6.50	0.111	0.303	0.094
Total slurry ^1^								
Weight, kg/cow/day	51.6 ^b^	53.1 ^b^	67.9 ^a^	62.6 ^ab^	5.48	0.043	0.038	0.070

SEM—standard error of least square means; ^ab^ Least square means in the same row with different superscript letters differ (*p* < 0.05); ^1^ Calculated as sum of weight of fresh faeces plus urine (per cow/day).

**Table 3 animals-12-01243-t003:** Effect of dietary CP level on NH_3_ emissions from slurry from lactating cows by measurement time interval.

	Dietary CP Level, g/kg DM					*p*-Value		
	141	151	177	201	SEM	Diet	Linear	Quadratic
0–24 h								
NH_3_, g/cow	21.3 ^c^	29.5 ^bc^	42.2 ^b^	63.4 ^a^	4.97	0.0002	<0.0001	0.538
24–48 h								
NH_3_, g/cow	9.4 ^c^	13.8 ^bc^	23.5 ^ab^	33.1 ^a^	3.80	0.003	0.0003	0.992
0–48 h ^1^								
NH_3_, g/cow	30.6 ^c^	43.4 ^bc^	65.8 ^b^	96.6 ^a^	7.91	0.0002	<0.0001	0.712
Average daily emissions								
NH_3_, g/cow/day	15.3 ^c^	21.7 ^bc^	32.9 ^b^	48.3 ^a^	3.96	0.0002	<0.0001	0.712

SEM—standard error of least square means; ^abc^ Least square means in the same row with different superscript letters differ (*p* < 0.05); ^1^ Total cumulative NH_3_ emissions during the entire study period.

**Table 4 animals-12-01243-t004:** Effect of chamber temperature on NH_3_ emissions from slurry from lactating cows by measurement time interval.

	Chamber Temperature, °C			
	8	18	SEM	*p*-Value
Measurement time interval				
0–24 h				
NH_3_, g/cow	33.9 ^b^	45.4 ^a^	4.78	0.009
24–48 h				
NH_3_, g/cow	29.4 ^a^	14.3 ^b^	3.20	0.001
0–48 h				
NH_3_, g/cow	63.4	59.7	6.56	0.475
Average daily emissions				
NH_3_, g/cow/day	31.7	29.8	3.28	0.475

SEM—standard error of least square means; ^ab^ Least square means in the same row with different superscript letters differ (*p* < 0.05).

**Table 5 animals-12-01243-t005:** Linear regression of NH_3_ emissions from slurry from lactating cows with dietary CP level, total urinary nitrogen (UN), and faeces N (FN).

Linear Equation								R^2^
NH_3_, g/Cow/Day =								
(1)	−60.09 (11.1)	+	0.54 (0.07) × CP ^a^					0.67
(2)	−30.98 (8.86)	+	6.55 (0.91) × UN ^b^					0.62
(3)	+30.04 (19.4)	+	0.32 (3.77) × FN ^c^					0.0002
(4)	−56.07 (10.9)	+	0.37 (0.11) × CP ^a^	+	2.66 (1.44) × UN ^b^			0.72
(5)	−47.2 (14.0)	+	0.56 (0.06) × CP ^a^	−	3.13 (2.12) × FN ^c^			0.71
(6)	−31.7 (15.0)	+	6.55 (0.92) × UN ^b^	+	0.14 (2.36) × FN ^c^			0.62
(7)	−47.4 (13.7)	+	0.41 (0.12) × CP ^a^	+	2.24 (1.49) × UN ^b^	−	2.25 (2.15) × FN ^c^	0.73

^a^ CP in g/kg DM; ^b^ UN in g/L; ^c^ FN in g/kg fresh weight.

## Data Availability

All data used in this paper were stored in the server of the Agri-Food and Biosciences Institute which can be accessed upon the request.

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
