# Peer review of "The Effects of Dietary Crude Protein Level on Ammonia Emissions from Slurry from Lactating Holstein-Friesian Cows as Measured in Open-Circuit Respiration Chambers"

_animals, 2022, doi:10.3390/ani12101243_

Round 1

Reviewer 1 Report

ID: animals-1677210

Title: The effects of dietary crude protein level on ammonia emissions from slurry from lactating Holstein-Friesian cows as measured in open-circuit respiration chambers

The objective of this study was to determine the effect of increasing dietary crude protein levels on slurry NH3 emissions from lactating Holstein-Friesian cows as measured in open-circuit respiration chambers. Although the work is interesting, some concerning points are shown below;

Abstract:

-Add a summary of the NH3 emission prediction equation.

Introduction:

-Provide some context for the livestock (ruminant) production situation in terms of greenhouse gas emissions and environmental impacts.

-Express some background on the relationship between N excretion and dietary protein intake. (The effect of protein digestion on N excretion).

-There is a lack of CP level documentation and review in order to support your objective.

Materials and Methods:

L77.

-How much does the experimental cow weigh, and how much milk does she consume per day?

L129-130.

-Why did the authors determine the urine-to-feces production in different ratios for each treatment diet?/ Why not use the same ratio?/ Are there any references or sources?

Slurry allocation to respiration chambers and NH3 emission measurements:

-What is the model of open-circuit respiration chambers? (Branding/model/chamber size/produce location)

-How much is the flow rate recorded by the flow meter in the chambers?

-What is the gas analyzer or detector of the chamber system in this study? (Branding/mode/produce location)

-How to measure fecal and urinary NH3?/ Please explain more details.

L161-16.

-In this study, it not reported the fecal DM, NDF, ADF.

-Should be explained about feed collection and feed chemical analysis too/ Add a reference.

-How do you examine fecal and urinary N? What methods are used for nitrogen analysis? (Include a reference and explain further)

Statistical Analysis:

-What’s the experimental design of this study?/ Please show the statistical model.

L182-193:

-Why do the authors have to compare the time at 0-24 hr with 0-48 hr or the average daily NH3?

-What else can knowledge be gained by comparing 0-24 and 0-48 hr? It makes sense to understand. If the authors need to compare the time at 0-24 and 24-48 hr. to report the different bioactivity for NH3 emissions after 24 hr. But 0-48 hr, we well know, is greater than 0-24 hr. 

Results and Discussions:

- In describing the experimental results for each topic. The authors should introduce the topic before describing it in each part, table, or figure. For example:

"Effect of dietary CP level on weights and N composition of the urine and faeces were shown in table2. …" 

L199:

-How does higher protein intake affect increased urinary excretion of urea?/ Please elaborate on the bio-mechanism.

L204:

-"The amount and N content of faeces did not differ.." Is it similar to other studies or not?/ Add reference

L206:

-"…showing a tendency (p = 0.070)…"/ It should be further stated in the statistical analysis section that the tendency value is expressed with 0.05≤P<0.10

L236:

-Where does the urease enzyme come from? / From where?

 L239:

-Is it need to be reduced to what level to be suitable without negative effects on animals and the environment?

L294:

-"…(p = 0.078)…" ? In table 4. was shown p=0.092 forhe average daily NH3 emissions

L298:

-Why does the evaporation of water increase NH3 emissions? (Please explain more about the mechanism)/ Or is there a possibility that higher temperatures may accelerate the activity of the urease enzyme?/ If so, please elaborate on the biomechanics.

L339.

-"Slopes of the regression equations for predicting NH3 emissions were not different…"/ In the experimental result, it wasn't reported or demonstrated that the comparison in the slopes of each equation was made.

-Why was the relationship between NH3 emissions and feces N weak and negative? /Please explain more about the mechanism.

Conclusions:

-Add a summary of the prediction equation.

-What happens to the prediction regression equation?

-What are suitable situations for using this prediction equation?

-What protein levels in feed are appropriate for cattle feeding with the least negative impact on the animal and the environment?.

Table1.

-In dietary 201 g CP, the R:C ratio is not included as 100% (47.5+52.3= 99.8)/ Please check!

-"DM, CP, NDF, ADF, WSC" should be using full words or add the explanation in the footnote.

Table2.

-Should be chosen to display the same decimal places (2 or 3 places) for SEM and p-value, for example: 0.527 change to 0.53/ 0.039 change to 0.04/ 0.006 change to <0.01/ <0.0001 change to <0.01 etc.

-The unit of urinary N, fecal N, and total N should be expressed in g/day.

Table3-6.

-Should be show p-value data in all parameters.

Table3.

-Should be chosen to display the same decimal places (2 or 3 places) for SEM and p-value.

-The average daily NH3  should be 17.95 g/cow/day or not?  (35.9÷ 2(day)=17.95)/ Please check.

Table5.

-Add FN description in the footnote.

-Why did the (3) equation have a too low R2-value?

Figure1.

-In chamber run 8, the RC at 8oC is not related to the description (It should be RC1,2,3 right?)

Author Response

The response is uploaded as a Word file.

Reviewer 2 Report

As a paper, I think it is well written, but I have several questions.

The novelty of this paper is unclear. All in all, there are no "surprises".
NH4+ --- "+" to superscript.

L49; NH3 --- subscript
Table 1. 141 --- Why only here in Bold?
TMR, NDF, ADF, WSC etc. --- Don't you need to explain the abbreviation?
Figure 1. Chamber run 8 Chamber temperature (8 oC) --- Where are cow numbers ?
L188; Did you not use the multiple comparison method?
L220; but not different between the 141, and 177 diets --- I cannot understand.
L265; [40], --- Is the underline in the , ?
Figure 2 --- Contents duplicate those in Table 3. Not required.
L296; NH3 --- subscript
Table 4. 18 --- Bold.
Figure 3 --- Contents duplicate those in Table 4. Not required.
L345, L347; NH3 --- subscript
Table 5. What is c ?
What are the numbers in ()?
L365; When will the other 45% be gone?

L199, L232, L241, L275 --- You use many "surprise". Since this is a scientific paper, no emotional expression is required.

Author Response

(The authors gave the same response as above.)

Round 2

Reviewer 1 Report

Journal: animals-1677210

Title: The effects of dietary crude protein level on ammonia emissions from slurry from lactating Holstein-Friesian cows as measured in open-circuit respiration chambers

Authors: Constantine Bakyusa Katongole , Tianhai Yan *

The second round of review:

Thank you for your thoughtful response to my prior suggestions; the paper has been revised. There are a few minor remarks that need to be revised by the writers. Please note the following particular comment:

-L16-17: Changed from "The influence of dietary crude protein (CP) levels on ammonia (NH3) emissions from lactating Holstein-Friesian cow slurry was investigated" to "The influence of dietary crude protein (CP) levels on ammonia (NH3) emissions from lactating Holstein-Friesian cow slurry was investigated."

-Provide a conclusion relating to the objective research at the end of the section. The CP level should be summarized.

Introduction:

-There is no literature review pertaining to the objective research. The level of CP should reflect how involved you are in the current study. When discussing the prior study's CP level, what is the research gap, and what has to be studied further?

-The research hypotheses are lacking and should be included.

Materials and Methods:

-L97-98: It is still unclear about the varied CP. How did the writers create this range?

L100-102: How do the writers eat? ad infinitum? Please specify.

Reviewer 2 Report

My comments are as follows,

L171; United Kingdom --- UK
L173; m3 --- Superscript
L178; m2 --- Superscript
L198; m2 --- Superscript
